# Implementation of point-of-care ultrasound in the medical intensive care unit: A retrospective analysis of physician practices and patient outcomes

Sam Tierney[1], Stephen Germana[2], Frank Papik[3], Werda Alam[4], Abigail Chua[1,2], Sunil Chulani[2], Karin Hasegawa[5], Xiaoyue Zhang[6], Sahar Ahmad[1,2]*

1 Renaissance School of Medicine at Stony Brook University, Stony Brook, New York, United States of America, 2 Department of Medicine, Stony Brook University Hospital, Stony Brook, New York, United States of America, 3 Department of Medicine, Oconee Memorial Hospital, Seneca, South Carolina, United States of America, 4 Department of Medicine, Mount Sinai West, New York, New York, United States of America, 5 Department of Applied Mathematics and Statistics, Stony Brook University Hospital, Stony Brook, New York, United States of America, 6 Biostatistical Consulting Core, Office of Scientific Affairs, Renaissance School of Medicine, Stony Brook University, Stony Brook, New York, United States of America

* Sahar.Ahmad@stonybrookmedicine.edu

## Abstract

### Background

Point of care ultrasound (POCUS) use has become ubiquitous in the ICU setting. Despite this, there remains a paucity of data regarding how its use impacts patient outcomes. We sought to determine if physician attitudes toward and use of POCUS impact patient outcomes.

### Objectives

To evaluate the impact of physician use of POCUS in the medical intensive care unit (MICU) on patient care outcomes.

### Methods

A longitudinal, single-center retrospective study of adult patients in a tertiary care center MICU was conducted between 2016 and 2017. A total of 512 patient encounters were analyzed for: number of ventilator days, number of vasopressor days, and MICU length of stay. The outcomes were then compared between attending physicians who showed high, medium, and low self-reported POCUS implementation and confidence scores by questionnaire.

**Data availability statement:** All relevant data are within the paper and its Supporting Information files.

**Funding:** The author(s) received no specific funding for this work.

**Competing interests:** The authors have declared that no competing interests exist.

## Results

Patients of physicians in the low POCUS confidence group had a statistically significant increase in MICU length of stay (IRR = 1.19, 95% CI: 1.06–1.32, p = 0.0022) and ventilator days (IRR = 1.18, 95% CI: 1.03–1.36, p = 0.0197) compared to the medium confidence group.

## Conclusions

Our investigation found a statistically significant relationship between self-reported confidence in POCUS by MICU physicians and MICU length of stay as well as total days spent on mechanical ventilation. Increased self-reported use of POCUS by physicians did not impact patient outcomes in this population. There was similarly no evidence of increased harm or adverse outcomes in high-POCUS implementation groups.

## Introduction

Point of care ultrasonography (POCUS) has emerged as the standard of care for many diagnostic and interventional purposes in the medical intensive care unit (MICU) and in the hospital setting generally. Its use has become more prevalent in the past 20 years, with a push by some for associated hospital credentialing [1,2]. The role of POCUS ranges from central line placement to echocardiogram, from thoracentesis to evaluation of pneumothorax and its use is ubiquitous in intensive care [3,4].

The role of POCUS is not merely diagnostic; its use has helped to improve the safety of various interventions [5]. The Society of Critical Care Medicine has made ultrasound guidance a 1A recommendation for central venous catheter placement. The use of this imaging modality shortens procedure time, improves success rate, and reduces risk of procedure-related complications [6]. A randomized control trial showed that ultrasound guided subclavian vein cannulation was successful in 100% of cases, compared to 87.5% with landmark technique alone. The ultrasound-guided group also had a decrease in complications including arterial puncture, hematomas, pneumothorax, hemothorax, phrenic nerve injury, and cardiac tamponade [7].

In recent years, data has emerged suggesting a role for POCUS in evaluating COVID-19. A study by Lichter *et al.* in 2020 showed that higher lung ultrasound scores strongly correlate to eventual need for mechanical ventilation and can be a strong predictor of mortality [8]. A similar study by Bonadia *et al.* showed that lung US can also be used to help prognosticate patients presenting to the emergency department [9]. A prospective cohort study by Kumar and colleagues found that abnormal findings on POCUS (e.g., consolidation, B-lines) in patients with COVID-19 was predictive of ICU admission and need for supplemental oxygen following discharge [10]. Such knowledge can be used to guide treatment strategies and prognosticate more accurately. POCUS is particularly useful in the setting of acute respiratory distress

syndrome (ARDS). In a prospective observational study, POCUS had a specificity of 97.7% for diagnosis of alveolar inter-stitial syndromes such as ARDS [11,12].

Despite the wealth of evidence that POCUS can be used as an effective clinical tool, there is a relative paucity of data regarding its impact, if any, on hard patient outcomes. Feng *et al*. analyzed the Medical Information Mart for Intensive Care (MIMIC-III) database and reviewed the use of transthoracic echocardiography (TTE) in the ICU. This analysis found that in patients with sepsis, use of TTE was associated with an improvement in 28-day mortality [13]. Other studies have shown that in hemodynamically unstable patients, early diagnosis of tamponade was made via ultrasound which expe-dited bedside pericardiocentesis [14,15]. A meta-analysis of RCTs found that POCUS use may reduce 28-day mortality, duration of vasopressor therapy, and need for renal replacement therapy in adult patients with shock, but was unable to investigate the impact of operator competency and POCUS timing on patient outcomes [16]. Interestingly, POCUS use has also been associated with a reduction in hospitalization cost and chest X-rays for hospitalized patients [17]. The impact on POCUS use on the outcomes in the MICU across diverse disease states, however, has yet to be fully eluci-dated and is an active area of research.

With this in mind, we sought to determine if MICU physicians' implementation of or confidence in the use of POCUS impacts patient outcomes. In this single-center retrospective study, we evaluated the relationship between physician attitudes towards POCUS, their resulting self-reported use of POCUS, and hard patient outcomes such as MICU length of stay (LOS), days spent on mechanical ventilation, and days spent on vasopressor therapy. It was our hypothesis that physicians who more heavily utilize POCUS in their daily practice will have patients who spend fewer days in the MICU, on ventilators, or being administered vasopressors on average. In general, the role of POCUS is discussed in the litera-ture for specific patient presentations, such as in sepsis or pneumonia. But aggregate outcomes following POCUS evalu-ation across the wide variety of pathologies that may present to a particular MICU are not well studied or understood. This analysis seeks to further elucidate that relationship.

## Methods

### Study cohorts

We conducted a single-center retrospective study using a previously established convenience sample database of all adult patients admitted to the MICU at Stony Brook University Hospital from November 2016 to April 2017. The Stony Brook University Office of Research Compliance reviewed the protocol and determined that the project qualified for exemption under federal exemption category 45 CFR 46.104(d) and issued the corresponding approvals to conduct the study (IRB# 2024−00109). During the study period, twelve attending MICU physicians were identified as having led the care of these patients. These physicians were administered a questionnaire after data acquisition was completed assess-ing their previous training in POCUS use, subjective confidence in the diagnostic power of POCUS in various acute pathologies, and degree of POCUS implementation in their own practice (S1 Text). Survey results were collected either by online questionnaire utilizing Qualtrics software ($n = 2$) or in-person interview ($n = 10$). All responses were subsequently coded to ensure physician privacy.

Based on these survey results, physicians were sorted into low-, medium- and high-POCUS implementation and con-fidence groups. For implementation groups, cutoffs were as follows: low, less than or equal to three uses of POCUS per day; medium, between four and seven uses per day; high, greater than or equal to eight uses per day. Physician POCUS confidence groups were determined based on survey data which indicated each physician's personal confidence in the utility and efficacy of POCUS for a list of pathologies in a score 18–90. For utility groups, cutoffs were as follows: low, a survey score of 66 or less; medium, a survey score 67–72; high, a survey score of 73 or greater. Similarly, for efficacy groups, cutoffs were as follows: low, a survey score of 71 or less; medium, a survey score 72–76; high, a survey score of 77 or greater. Score cutoffs between physician groups were chosen to ensure roughly even distribution of the twelve physicians between groups. Differences between efficacy and utility groups were minor and ultimately had no impact on

the observed results; therefore, "confidence" is used here as an umbrella term to reflect both utility and efficacy physician groups. For each group, average number of patient-days (i.e., the number of days that each physician group spent caring for patients, on average) was calculated to ensure similar degree of patient exposure between physician groups.

The database sample of MICU admissions initially included 1,069 patients. Of these, 557 were excluded due to missing data, leaving 512 patients that qualified for analysis. For each of these, age; sex; in-unit mortality; MICU LOS; days spent on mechanical ventilation; days spent on vasopressor therapy; and length of time spent under the care of each attending physician was collected via electronic health record (EHR) review (S2 Dataset). Patient data was accessed for research purposes on August 7, 2024 and while identifying information for patients was initially accessed for the purposes of collecting all pertinent data, it was subsequently anonymized. The original data, which was compiled in a patient-based table (i.e., each row corresponds to unique patient information), was then transformed to a physician-based data set, where each entry corresponds to a unique physician-patient combination. Therefore, for those patients who were seen by more than one physician, multiple entries are allowed resulting in an overall patient number, N, greater than the number of total patient encounters. The physicians were then grouped into POCUS implementation and confidence groups.

### Patient outcomes

Patient outcomes for this study were days spent on mechanical ventilation, days spent on vasopressor therapy, and MICU LOS. Available records from all the 512 patients were included in the MICU LOS analysis. Analyses of ventilation and vasopressor days were limited to patients who received those respective therapies (i.e., ventilation days > 0 or vasopressor use > 0).

### Statistical analysis

The chi-squared test was used to examine the marginal association between a categorical variable (i.e., patient sex) and physician ultrasound variables (i.e., implementation and confidence). The Kruskal-Wallis test was used to examine the association between the numerical variables (i.e., age, APACHE III score and number of patient-days) and the physician ultrasound variables, as well as the association between the physician ultrasound variables and the continuous outcomes (i.e., MICU LOS, ventilator days, vasopressor days).

Following the above univariate analyses, generalized linear regression models with negative binomial distribution and log link were then utilized to model the relationship between each outcome and each physician's ultrasound variable with covariates. The estimated model parameters were exponentiated and expressed as incidence rate ratio (IRR). An IRR > 1 indicated a longer length of stay or more days on therapy, and an IRR < 1 indicated a shorter duration. P-values less than 0.05 were considered statistically significant. All the statistical analysis was performed using SAS 9.4 (SAS Institute Inc., Cary, NC).

### Covariates

Patients' age, gender, and severity of presentation at time of admission to the MICU and physicians' number of patient-days were all considered covariates for the purpose of this analysis. Presentation severity was measured by APACHE III score [18].

### Results

From univariate analysis, patients' demographics as well as the average patient-days were not significantly different across all three levels of either POCUS implementation or confidence (Table 1). This indicates that the twelve physicians participating in this study, despite varying levels of POCUS implementation and confidence, were treating similar patient populations for similar periods of time. The main covariate considered in this study was patients' initial clinical presentation severity as represented by APACHE III score. Much like other parameters, average APACHE III was not

**Table 1. Patients' demographics and physician average patient-day stratified by POCUS implementation and confidence levels.**

| Implementation | | | | | |
|---|---|---|---|---|---|
| | Overall (*N*=836) | Low (*N*=203) | Medium (*N*=313) | High (*N*=320) | *p* |
| Age | 68±23 | 69±25 | 68±23 | 69±24 | 0.6328 |
| Sex (% Female) | 45.57% | 48.28% | 47.28% | 42.19% | 0.2943 |
| APACHE III Score | 61.5±34 | 61±34 | 61±32 | 63.5±37 | 0.7636 |
| Patient Days | 3±2.5 | 3±4 | 3±3 | 3±2 | 0.9384 |
| Confidence | | | | | |
| | Overall (*N*=836) | Low (N=250) | Medium (N=254) | High (N=332) | *p* |
| Age | 68±23 | 68±23 | 67±24 | 69±24 | 0.4469 |
| Sex (% Female) | 45.57% | 43.6% | 48.81% | 44.58% | 0.4485 |
| APACHE III Score | 61.5±34 | 61±31 | 61±37 | 62±35 | 0.9178 |
| Patient Days | 3±2.5 | 3±3 | 3±3 | 3±2 | 0.7868 |

Note 1: Some patients were seen by more than one physician and therefore counted more than once, resulting in a higher number of records than unique patients included in the study.

Note 2: For categorical variables, column percentages across different levels of a physician's POCUS levels and p-values based on Chi-square test were reported. For continuous variables, median and IQR, and non-parametric p-values based on Kruskal-Wallis test were reported.

significantly different across physician POCUS implementation levels (Table 1, *p*=0.7636): 61 (±34) in the low, 61 (±32) in the medium, and 63.5 (±37) in the high implementation groups. Similarly, average APACHE III score was not significantly different across physician POCUS confidence levels (S1 Table, p=0.9178). This also indicates no statistically significant difference across physician implementation and confidence levels in terms of patient presentation severity that could account for potential differences in measured outcomes.

The level of POCUS training did vary in physicians across implementation and confidence levels, however. In the high POCUS implementation group, 66% of the physicians had some degree of formal training in POCUS, compared to 80% and 25% in the medium- and low-implementation groups, respectively. Training differences were similarly stratified in POCUS confidence groups (75%, 75%, and 25% in high-, medium-, and low-confidence groups, respectively). The degree of previous training was determined based on survey results (S3 Dataset).

Univariate analyses revealed no statistically significant differences in distribution of patients' MICU LOS, days on mechanical ventilation, or days on vasopressor therapy across three levels of POCUS implementation, confidence (S1-S3 Tables). Median MICU LOS was 5 days for all three implementation levels (S1 Table. p=0.8558) with IQR of 6, 5, and 5 days for high, medium, and low levels, respectively. Of the three levels of POCUS confidence, median MICU LOS was 5 days (S1 Table, p=0.0751) with IQR of 4, 5, and 6 days for high, medium, and low levels, respectively. Patients who underwent invasive mechanical ventilation were on ventilation for median of 2, 2 and 3 days (S2 Table, p=0.7579) with IQR of 3, 3 and 4 days for high, medium and low implementation levels, respectively, and for median of 2, 3 and 3 days (S2 Table , p=0.2496, IQR=3 days for all) for high, medium and low confidence levels, respectively. Similarly, patients under the care of high, medium, and low POCUS implementation physicians who were on vasopressor therapy all had median length of vasopressor treatment of 2 days with IQR of 2, 3 and 2 days for high, medium and low levels, respectively (S3 Table , *p*=0.6574) while patients had median length of vasopressor treatment of 2 days for all three confidence levels (S3 Table , *p*=0.4333) with IQR of 2, 3 and 2 days for high, medium and low levels, respectively.

Multivariable regression analyses showed that there was a statistically significant difference in the patients' MICU length of stay among the different levels of physicians' ultrasound confidence (Table 2, p=0.0089). Patients who were seen by the physicians whose ultrasound confidence level were low had 19% longer MICU length of stay compared to those patients who were seen by the physicians whose confidence level were medium (Table 2, IRR=1.19, 95% CI: 1.06–1.32,

**Table 2. Incidence rate ratio (IRR) and 95% CI of explanatory variables for MICU length of stay based on negative binomial regression models.**

| Outcome | Variable | Effect | Level | IRR and 95% CI | P-value* | P-value** |
|---|---|---|---|---|---|---|
| MICU Length of Stay | Implementation | Age | Unit=1 | 1.0006 (0.9980, 1.0033) | 0.6422 | 0.6422 |
| | | APACHE III | Unit=1 | 1.0042 (1.0025, 1.0060) | <.0001 | <.0001 |
| | | Patient Days | Unit=1 | 1.17 (1.15, 1.19) | <.0001 | <.0001 |
| | | Sex | Female vs Male | 1.19 (1.09, 1.30) | <.0001 | <.0001 |
| | | Implementation | High vs Low | 1.08 (0.97, 1.21) | 0.1507 | 0.2528 |
| | | | High vs Medium | 1.07 (0.97, 1.18) | 0.1709 | |
| | | | Low vs Medium | 0.99 (0.88, 1.10) | 0.8208 | |
| | Confidence | Age | Unit=1 | 1.0007 (0.9980, 1.0034) | 0.6180 | 0.6180 |
| | | APACHE III | Unit=1 | 1.0043 (1.0025, 1.0060) | <.0001 | <.0001 |
| | | Patient Days | Unit=1 | 1.17 (1.15, 1.19) | <.0001 | <.0001 |
| | | Sex | Female vs Male | 1.19 (1.10, 1.30) | <.0001 | <.0001 |
| | | Confidence | High vs Low | 0.91 (0.82, 1.01) | 0.0639 | 0.0089 |
| | | | High vs Medium | 1.08 (0.97, 1.19) | 0.1603 | |
| | | | Low vs Medium | 1.19 (1.06, 1.32) | 0.0022 | |

*: P-values were calculated based on type 3 T test from the negative binomial regression models.

**: P-values were calculated based on type 3 F test from the negative binomial regression models.

p = 0.0022). Similarly, multivariable regression analysis showed there was a statistically significant difference in the patients' ventilator days among the different levels of doctor's ultrasound confidence (Table 3, p = 0.0468), where those patients who were seen by the physicians whose ultrasound confidence were low had 18% longer days on ventilator compared to those patients who were seen by the physicians whose ultrasound confidence were medium (Table 3, IRR = 1.18, 95% CI: 1.03–1.36, p = 0.0197). The rest of the physicians' POCUS levels were not significantly associated with any of the patients' outcomes (Tables 2–4).

## Discussion

The results of this single-center retrospective analysis show a significant increase in MICU length of stay and days spent on mechanical ventilation for patients in the low POCUS physician confidence group. Interestingly, similar results were not seen in patient groups stratified by degree of physician POCUS utilization. The demographics of patients across physician groups confirms that the patient outcomes data has not been influenced by differences in patient populations (see Table 1). There was no significant difference in patient age or APACHE III scores across groups, indicating that each physician group was overseeing care for patients of similar average acuity. Furthermore, there was likewise no difference in the number of patient-days between physician groups. Taken as a whole, these demographic findings indicate that our results are not secondary to an inconsistency in patient populations and that our work might be a signal that use of POCUS in the MICU is correlated with improved patient outcomes. To our knowledge, this is the first study that has investigated hard MICU outcomes across a diverse patient population in relation to attending physician implementation of POCUS.

Assessing the clinical value of new diagnostic modalities can be difficult, especially in complex clinical situations. As the capabilities of medical technology continue to expand, it is important to assess the impact of these modalities on hard patient outcomes to investigate their diagnostic use [19]. Discourse within the medical literature about the clinical use of new modalities is to be expected. A key example of this is the decline in favor of pulmonary artery catheter use in recent years following evidence that frequent use does not improve patient outcomes [20–22]. Similarly, POCUS finds itself in a period of investigation of its impact on patient outcomes. There remains a lack of consensus in the literature as to the effect of routine POCUS use on hard patient outcomes in the MICU. Previous investigations have shown outcome benefits

**Table 3. Incidence rate ratio (IRR) and 95% CI of explanatory variables for days on mechanical ventilation based on negative binomial regression models.**

| Outcome | Variable | Effect | Level | IRR and 95% CI | P-value* | P-value** |
|---|---|---|---|---|---|---|
| Days on Ventilator | Implementation | Age | Unit=1 | 0.9979 (0.9944, 1.0014) | 0.2402 | 0.2402 |
| | | APACHE III | Unit=1 | 1.0022 (1.0002, 1.0042) | 0.0315 | 0.0315 |
| | | Patient Days | Unit=1 | 1.19 (1.18, 1.21) | <.0001 | <.0001 |
| | | Sex | Female vs Male | 1.1147 (0.9969, 1.2464) | 0.0563 | 0.0563 |
| | | Implementation | High vs Low | 1.09 (0.95, 1.25) | 0.2364 | 0.3912 |
| | | | High vs Medium | 1.08 (0.95, 1.22) | 0.2471 | |
| | | | Low vs Medium | 0.99 (0.86, 1.14) | 0.8944 | |
| | Confidence | Age | Unit=1 | 0.9981 (0.9945, 1.0016) | 0.2909 | 0.2909 |
| | | APACHE III | Unit=1 | 1.0022 (1.0002, 1.0042) | 0.0316 | 0.0316 |
| | | Patient Days | Unit=1 | 1.19 (1.18, 1.21) | <.0001 | <.0001 |
| | | Sex | Female vs Male | 1.1174 (0.9994, 1.2493) | 0.0509 | 0.0509 |
| | | Confidence | High vs Low | 0.8768 (0.7685, 1.0003) | 0.0506 | 0.0468 |
| | | | High vs Medium | 1.03 (0.90, 1.18) | 0.6201 | |
| | | | Low vs Medium | 1.18 (1.03, 1.36) | 0.0197 | |

*: P-values were calculated based on type 3 T test from the negative binomial regression models.

**: P-values were calculated based on type 3 F test from the negative binomial regression models.

**Table 4. Incidence rate ratio (IRR) and 95% CI of explanatory variables for days on vasopressor therapy based on negative binomial regression models.**

| Outcome | Variable | Effect | Level | IRR and 95% CI | P-value* | P-value** |
|---|---|---|---|---|---|---|
| Days on Pressor | Implementation | Age | Unit=1 | 0.9977 (0.9921, 1.0034) | 0.4360 | 0.4360 |
| | | APACHE III | Unit=1 | 1.0022 (0.9994, 1.0050) | 0.1221 | 0.1221 |
| | | Patient Days | Unit=1 | 1.14 (1.11, 1.17) | <.0001 | <.0001 |
| | | Sex | Female vs Male | 1.08 (0.91, 1.29) | 0.3775 | 0.3775 |
| | | Implementation | High vs Low | 1.02 (0.83, 1.27) | 0.8332 | 0.9297 |
| | | | High vs Medium | 0.98 (0.80, 1.20) | 0.8538 | |
| | | | Low vs Medium | 0.96 (0.77, 1.19) | 0.7028 | |
| | Confidence | Age | Unit=1 | 0.9977 (0.9920, 1.0034) | 0.4241 | 0.4241 |
| | | APACHE III | Unit=1 | 1.0021 (0.9993, 1.0049) | 0.1388 | 0.1388 |
| | | Patient Days | Unit=1 | 1.14 (1.11, 1.17) | <.0001 | <.0001 |
| | | Sex | Female vs Male | 1.08 (0.91, 1.29) | 0.3762 | 0.3762 |
| | | Confidence | High vs Low | 0.99 (0.80, 1.21) | 0.9093 | 0.9609 |
| | | | High vs Medium | 0.97 (0.79, 1.20) | 0.7781 | |
| | | | Low vs Medium | 0.98 (0.79, 1.22) | 0.8665 | |

*: P-values were calculated based on type 3 T test from the negative binomial regression models.

**: P-values were calculated based on type 3 F test from the negative binomial regression models.

for patients with certain presentations, such as sepsis and dyspnea [20,23]. However, others have shown no impact on patient outcomes, and there has been at least one example of emergency department POCUS use being associated with increased in-hospital mortality due to delays in patient care by POCUS assessment [24,25]. However, our analysis seems to indicate that POCUS may represent a clinical and diagnostic tool which has the capacity to improve patient outcomes in the MICU.

An interesting finding from our analysis is that prolonged MICU length of stay and ventilator days were only seen in the low POCUS confidence group, but not in the group of physicians who self-reported low POCUS use in their practice. More research will be required to elucidate the reasons for this, but it is possible that this finding is representative of operator skill, a potential limiting factor to POCUS utility. It is worth noting that, of those physicians in the low POCUS confidence group, only 25% had some degree of formal training in bedside POCUS. Physicians who lack formal training in POCUS and thus expressed low confidence in POCUS as a diagnostic modality on survey, may still utilize it in their patient care but may not have the knowledge base to apply it to a wider range of pathologies and clinical situations. Physicians who self-reported low POCUS utilization, however, may make use of other diagnostic modalities (e.g., CT) in lieu of ultrasound, and thus did not have a statistically significant change in their patient outcomes. Whether this difference in patient outcomes is clinically significant has yet to be determined.

Importantly, our analysis did not find any worsened patient outcomes in the high ultrasound confidence and implementation groups. In addition, we investigated only hard patient outcomes, and did not address soft outcomes, such as patient/family satisfaction or physician confidence in care rendered. Indeed, previous work has indicated that the use of POCUS is associated with socioemotional benefits such as improved patient satisfaction in their care, reassurance, and knowledge of their pathology [26,27]. The positive impact of these effects should not be underestimated in the evaluation of POCUS as a diagnostic modality.

Our analysis was limited largely due to study design, namely the retrospective design and reliance on physician self-reported POCUS use, the latter of which introduces a measure of recall bias. In addition, while our investigation utilized a large convenience sample of MICU patients, our sample size of physicians was small and may not be generalizable to a larger cohort. Lastly, these results may not be generalizable to other MICUs with different patient populations or availability of resources. Further investigations should be designed to reproduce these findings in a larger, more robust manner.

## Conclusions

Our investigation found a statistically significant relationship between self-reported confidence in POCUS by MICU physicians and MICU length of stay as well as ventilator days measured by retrospective review of the EHR. This analysis did not find any association between POCUS use and adverse patient outcomes. While more investigation is required to elucidate the relationship between POCUS use and patient outcomes, this study may signal that in the setting of ubiquitous use and perceived benefits, POCUS implementation in the MICU may be correlated with positive effects on key patient outcomes.

## Consent to participate

Requirement for informed consent was waived by the Institutional Review Board at Stony Brook University Hospital as analysis was done on anonymized data.

## Consent for publication

Written informed consent for publication of anonymized physician survey results was obtained from each physician that participated in this study.

## Supporting information

**S1 Table. Summary statistics of MICU length of stay across physician POCUS implementation and confidence group (all data).**
(DOCX)

**S2 Table. Summary statistics of vent days across physician POCUS implementation and confidence group (ventilator positive data).**
(DOCX)

**S3 Table. Summary statistics of pressor days across physician POCUS implementation and confidence group (vasopressor positive data).**
(DOCX)

**S1 Text. Physician POCUS Attitudes Survey.**
(DOCX)

**S2 Dataset. Raw patient data deidentified.**
(XLSX)

**S3 Dataset. Physician survey results deidentified.**
(XLSX)

## Acknowledgments

We acknowledge the biostatistical consultation and support provided by the Biostatistical Consulting Core at the Renaissance School of Medicine, Stony Brook University.

## Author contributions

**Conceptualization:** Frank Papik, Werda Alam, Abigail Chua, Sunil Chulani, Sahar Ahmad.

**Data curation:** Sam Tierney, Stephen Germana, Frank Papik, Werda Alam, Sunil Chulani.

**Formal analysis:** Sam Tierney, Stephen Germana, Karin Hasegawa, Xiaoyue Zhang.

**Investigation:** Sam Tierney, Stephen Germana.

**Methodology:** Frank Papik, Werda Alam, Abigail Chua, Sahar Ahmad.

**Project administration:** Sahar Ahmad.

**Supervision:** Sahar Ahmad.

**Writing – original draft:** Sam Tierney.

**Writing – review & editing:** Sam Tierney, Stephen Germana, Abigail Chua, Sahar Ahmad.

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
