## [Decision Letter · Decision Letter 0]

16 Jun 2025

Dear Dr. Tierney,

We look forward to receiving your revised manuscript.

Kind regards,

Hamidreza Badeli

Academic Editor

PLOS ONE

Journal Requirements:

Reviewers' comments:

Reviewer's Responses to Questions

**Comments to the Author**

1. Is the manuscript technically sound, and do the data support the conclusions?

Reviewer #1: Partly

Reviewer #2: Partly

2. Has the statistical analysis been performed appropriately and rigorously?

Reviewer #1: No

Reviewer #2: No

3. Have the authors made all data underlying the findings in their manuscript fully available?

Reviewer #1: Yes

Reviewer #2: Yes

4. Is the manuscript presented in an intelligible fashion and written in standard English?

Reviewer #1: No

Reviewer #2: Yes

Reviewer #1: 1. There are inconsistencies in the quality of writing throughout the manuscript. A comprehensive review and editing of English language is necessary to increase the quality and readability of the text, specially in certain parts including the abstract section. For example: "We sought to determine if physician attitudes toward and use of POCUS impact patient outcomes."

2. The literature review done for the paper seems inadequate. A rudimentary search in Pubmed brings a lot of recent studies within the same subject area including RCTs and Cohorts. However, the authors have mentioned a scant few of these studies. A better more robust literature review is warranted with proper citations.

NZ

3. The study design tries to merge subjective data with objective outcomes, but raises some questions. How are the cut-offs for stratifications chosen? Is it based on a specific literature or are they chosen arbitrarily? Also the paper does not clarify when the surveys were done? Were they done after the collection of data or during it?

3. Table 1 is reported in the methods section which is odd as it is usually a part of the results.

4. Statistical analysis done on the data needs to be more robust. Heavy reliance on p-value and mean as the only key measures is not acceptable and limits the ability to assess the precision of the estimates. Including confidence intervals and effect sizes would strengthen the reliability of the reported findings. Of course it is understandable that data such as APACHE III score cannot benefit from ES, but other reported outcomes can and should include ES.

Reviewer #2: This manuscript explores the relationship between MICU physician use of point-of-care ultrasound (POCUS) and patient outcomes, including ventilator days and MICU length of stay. As a pulmonologist, I appreciate the study's effort to assess whether POCUS, widely accepted as a useful bedside tool, actually translates into measurable clinical benefits.

1.Relying on self-reported physician use introduces a source of bias. it would be helpful to know whether these physicians were formally credentialed or trained in POCUS.

2. The use of appropriate effect sizes can help with better interpretation. Furthermore at least a post hoc power analysis is necessary to determine what this sample size can actually achieve.

3.The use of categorized “high,” “medium,” and “low” implementation and confidence groups is reasonable, but the authors should clarify how the cutoffs were justified. Please justify POCUS stratification thresholds that you have used.

4.Minor grammatical errors are present in the manuscript. A review and revision is necessary.

**Do you want your identity to be public for this peer review?** For information about this choice, including consent withdrawal, please see our Privacy Policy

Reviewer #1: No

Reviewer #2: No

---

## [Author Response · Author response to Decision Letter 1]

27 Jul 2025

We thank you for your thoughtful review of our manuscript, “Implementation of point-of-care ultrasound in the medical intensive care unit: A retrospective analysis of physician practices and patient outcomes”. We value your feedback and the opportunity to contribute to the field’s understanding of the utility of POCUS in the MICU. We have reviewed the reviewers’ comments and have made appropriate edits to the original version of the manuscript as detailed below. We hope that these changes are satisfactory and are to the standards required for publication within PLOS One.

1. ‘There are inconsistencies in the quality of writing throughout the manuscript. A comprehensive review and editing of English language is necessary to increase the quality and readability of the text, specially in certain parts including the abstract section. For example: "We sought to determine if physician attitudes toward and use of POCUS impact patient outcomes."’

‘Minor grammatical errors are present in the manuscript. A review and revision is necessary.’

We have extensively reviewed and edited the manuscript for grammatical errors and believe that all have been identified and corrected.

2. ‘The literature review done for the paper seems inadequate. A rudimentary search in Pubmed brings a lot of recent studies within the same subject area including RCTs and Cohorts. However, the authors have mentioned a scant few of these studies. A better more robust literature review is warranted with proper citations.’

The majority of literature regarding POCUS use has been focused on its utility as a diagnostic study. As such, there are relatively few high-power studies investigating the impact of POCUS use on endpoints such as length of hospital stay and patient mortality. However, this is an active area of investigation and our literature review has been expanded to reflect this. We have added studies by Kumar et al. (reference 12; manuscript pg 3, lines 25-27), Basmaji et al. (reference 16; manuscript pg 4, lines 8-11), and Tierney et al. (reference 17; manuscript pg 4, lines 11-12) to better reflect the work done in this area to date.

3. ‘The study design tries to merge subjective data with objective outcomes, but raises some questions. How are the cut-offs for stratifications chosen? Is it based on a specific literature or are they chosen arbitrarily? Also the paper does not clarify when the surveys were done? Were they done after the collection of data or during it?’

‘The use of categorized “high,” “medium,” and “low” implementation and confidence groups is reasonable, but the authors should clarify how the cutoffs were justified. Please justify POCUS stratification thresholds that you have used.’

Thank you for these points, we agree that this section of the manuscript requires clarification. We have updated it accordingly. Physician surveys were administered after data collection was completed (see updated manuscript; pg 5, line 1). In regards to the implementation and confidence groups, cutoffs were chosen to ensure an even distribution of the twelve physicians across the groups (see updated manuscript; pg 5, lines 13-14).

4. ‘Table 1 is reported in the methods section which is odd as it is usually a part of the results.’

Table 1 has been relocated from the Methods section to the Results section.

5. ‘Statistical analysis done on the data needs to be more robust. Heavy reliance on p-value and mean as the only key measures is not acceptable and limits the ability to assess the precision of the estimates. Including confidence intervals and effect sizes would strengthen the reliability of the reported findings. Of course it is understandable that data such as APACHE III score cannot benefit from ES, but other reported outcomes can and should include ES.’

‘The use of appropriate effect sizes can help with better interpretation. Furthermore at least a post hoc power analysis is necessary to determine what this sample size can actually achieve.’

We have completed additional statistical analysis which has significantly changed the findings of our study and have resulted in large scale edits to the manuscript. We completed a multivariable regression analysis of the data that allowed us to model the relationship between each outcome and physician implementation and confidence groups. This analysis showed a statistically significant increase in MICU length of stay and days spent on mechanical ventilation in the low POCUS confidence group compared to the medium confidence group. This has greatly changed the conclusions of our analysis and the manuscript has been updated throughout accordingly. See updated Methods and Results sections for further detail. In light of these changes, we no longer feel that a post hoc power analysis is indicated. All statistical analysis was performed by dedicated biomedical core statisticians at Stony Brook University.

6. ‘Relying on self-reported physician use introduces a source of bias. It would be helpful to know whether these physicians were formally credentialed or trained in POCUS.’

We agree that our study’s reliance on self-reported physician use introduces a not-insignificant source of bias (discussed on manuscript pg 9, lines 12-13). We further agree that adding additional information about level of POCUS training in the physicians who participated in this study will be worthwhile. We collected this information when the participants were initially interviewed and have added this information to the Supplementary Materials. In addition, we have added a brief discussion of the percentage of physicians across groups who have had formal POCUS training prior to the time period covered by the study (see updated manuscript; pg 7, lines 7-12).

Again, we thank you for your assistance in improving our manuscript. We hope that the changes detailed above meet the standards of publication expected by PLOS One.

---

## [Editor Report · Decision Letter 1]

6 Aug 2025

Implementation of point-of-care ultrasound in the medical intensive care unit: A retrospective analysis of physician practices and patient outcomes

PONE-D-25-12695R1

Dear Dr. Tierney,

We’re pleased to inform you that your manuscript has been judged scientifically suitable for publication and will be formally accepted for publication once it meets all outstanding technical requirements.

Kind regards,

Afagh Hassanzadeh Rad

Academic Editor

PLOS ONE
---

## [Editor Report · Acceptance letter]

PONE-D-25-12695R1

PLOS ONE

Dear Dr. Tierney,

I'm pleased to inform you that your manuscript has been deemed suitable for publication in PLOS ONE. Congratulations! Your manuscript is now being handed over to our production team.

Kind regards,

on behalf of

Dr. Afagh Hassanzadeh Rad

Academic Editor

PLOS ONE